# Boosting biodiversity monitoring using smartphone-driven, rapidly accumulating community-sourced data

**Keisuke Atsumi[1]\*, Yuusuke Nishida[1], Masayuki Ushio[2,3,4], Hirotaka Nishi[5], Takanori Genroku[1], Shogoro Fujiki[1,4]\***

[1]Biome Inc, Kyoto, Japan; [2]Department of Ocean Science, Hong Kong University of Science and Technology, Kowloon, Hong Kong; [3]Hakubi Center, Kyoto University, Kyoto, Japan; [4]Center for Ecological Research, Kyoto University, Shiga, Japan; [5]Toyohashi Museum of Natural History, Aichi, Japan

**Abstract** Comprehensive biodiversity data is crucial for ecosystem protection. The *Biome* mobile app, launched in Japan, efficiently gathers species observations from the public using species identification algorithms and gamification elements. The app has amassed >6 million observations since 2019. Nonetheless, community-sourced data may exhibit spatial and taxonomic biases. Species distribution models (SDMs) estimate species distribution while accommodating such bias. Here, we investigated the quality of *Biome* data and its impact on SDM performance. Species identification accuracy exceeds 95% for birds, reptiles, mammals, and amphibians, but seed plants, molluscs, and fishes scored below 90%. Our SDMs for 132 terrestrial plants and animals across Japan revealed that incorporating *Biome* data into traditional survey data improved accuracy. For endangered species, traditional survey data required >2000 records for accurate models (Boyce index ≥ 0.9), while blending the two data sources reduced this to around 300. The uniform coverage of urban-natural gradients by *Biome* data, compared to traditional data biased towards natural areas, may explain this improvement. Combining multiple data sources better estimates species distributions, aiding in protected area designation and ecosystem service assessment. Establishing a platform for accumulating community-sourced distribution data will contribute to conserving and monitoring natural ecosystems.

## eLife assessment

This **important** study presents findings of great practical value, offering fresh insights into natural species distributions across Japan. By combining multiple data sources (including those from non-academic sectors, aka citizen scientists), the manuscript also presents a **compelling** new tool that can be used to aid conservation agendas, detect species distribution changes, and testing of ecological theories.

## Introduction

Nature underpins human society, and the conservation of ecosystems and associated ecosystem services contributes to the sustainable development of human society, yet these services have been rapidly declining in recent years (***IPBES, 2019***; ***Loh et al., 2005***; ***Newbold et al., 2016***; ***Scholes and Biggs, 2005***). The Kunming-Montreal Global Biodiversity Framework (KM-GBF) by the United Nations envisions reversing the nature loss by 2030. As direct means for nature conservation, KM-GBF targeted making 30% of Earth's land and ocean area as protected areas by 2030 (i.e. 30 by 30). As

**\*For correspondence:**
k.atsumi115@gmail.com (KA);
fujiki@biome.co.jp (SF)

**eLife digest** The internet has allowed people to share their experiences through images, videos or audio recordings. This has led to the creation of online communities around a variety of topics, including biodiversity. In 2019, a smartphone app, called Biome, was created to fuel biodiversity engagement by making wildlife surveying an easy and fun activity via gamification and assisted species identification through image recognition and ecological analyses.

These types of observations are essential for understanding biological communities and species habitats, and they can indicate where and when species occur. Across Japan, Biome has gathered over 6.5 million observations of different species. For biologists, this type of data is extremely useful because it is continuous and enables advanced statistical estimations of species distributions. The fact that the approach is enjoyable to the user also means more people are willing to participate, lowering the barriers to collecting data about biodiversity loss.

However, questions remain regarding whether community-sourced data is robust enough for scientific purposes. To address this, Atsumi et al. investigated the quality of occurrence data collected in Biome. The researchers found that community identification of birds, reptiles, mammals and amphibians all exceeded 95% in accuracy. However, the accuracy fell for harder-to-judge seed plants, molluscs and fish species, ranging below 90%.

Atsumi et al. also compared how estimated distributions of each species changed when only scientific data was used, versus when it was combined with community data. To perform this analysis, the scientists recognized variations in observation efforts across different locations and individuals and adjusted for these biases in their estimations. They found that adding community-sourced data significantly improved the accuracy of species distribution estimations, including endangered species.

Atsumi et al. demonstrate that Biome data is useful when deciding which areas to designate as protected in terms of biodiversity. Additionally, these data can provide guidance for stakeholder-informed ecosystem service assessments. The element of rapid and reliable data collection can contribute to growing positive attitudes towards nature and biodiversity, The platform's community-driven nature also indicates an increase in biodiversity awareness and may link to crafting informative socio-environmental policy commitments.

an indirect but influential way, KM-GBF requires companies to "monitor, assess, and transparently disclose their risks, dependencies and impacts on biodiversity through their operations, supply and value chains and portfolios," which is guided by the Taskforce on Nature-related Financial Disclosures (TNFD) (*TNFD, 2023*). To achieve these goals, it is imperative to assess the state of biodiversity with a sufficient spatiotemporal resolution to support conservation planning, adaptive management, and companies' annual nature-related financial disclosures. The basis for such assessments lies in our knowledge of species distributions (*Gonzalez et al., 2023*; *Newbold et al., 2016*). Traditionally, distribution data was acquired through on-site surveys by experts (people have expertise about biodiversity), but collecting distribution data with sufficient spatiotemporal resolution is challenging if we rely only on such limited human resources (*Miya et al., 2022*; *Mori et al., 2023*; *Pocock et al., 2018*).

Since the emergence of digital devices and the internet, people have been able to share their observations through various media, such as images and video/audio recordings. Such community-sourced data have significantly contributed to the accumulation of ecosystem information. These datasets have been instrumental in assessing the impacts of climate change and urbanisation on phenology (*Fuccillo Battle et al., 2022*; *Klinger et al., 2023*), detecting distribution changes including invasive alien species (*Larson et al., 2020*; *Roy et al., 2023*; *Wallace and Bargeron, 2014*), exploring large-scale geographic variations in traits (*Atsumi and Koizumi, 2017*; *Leighton et al., 2016*), and estimating species distributions (*Chandler et al., 2017*; *Feldman et al., 2021*; *Johnston et al., 2018*; *Steen et al., 2019*). Moreover, the utilisation of machine learning to describe population trends based on community-sourced data (*Fink et al., 2023*) offers opportunities for conducting time-series analyses. These analyses can help us understand community assembly processes, unravel species interaction networks, and assess ecosystem stability (*Cornwell and Ackerly, 2009*; *Tilman et al., 2006*; *Ushio et al., 2018*), capitalising on the spatiotemporally dense sampling effort facilitated by community-sourced data (*Chandler et al., 2017*; *Kobori et al., 2016*; *Pocock et al., 2017*). Such analytical

approaches enable us to make informed predictions about changes in species distribution, population dynamics, and ecosystem stability in the face of climate change (*Bury et al., 2021*; *Pennekamp et al., 2019*; *Urban et al., 2016*). In essence, community-sourced data, owing to its extensive sampling across time and space, has the potential to test existing ecological theories, expand our comprehension of ecosystems and the underlying processes, eventually allowing us to forecast ecological dynamics in the context of climate change.

When people photograph organisms using digital devices with GPS capabilities, the images often contain timestamps and location details. Such images, when accompanied by species identifications, serve as evidence for tracking phenology and species occurrences. This crowdsourcing approach has been particularly successful on web- or mobile-based platforms such as eBird and iNaturalist (*Chandler et al., 2017*; *Wood et al., 2011*). Individuals submit records to these platforms for various reasons, including a desire to contribute to science and engage with cutting-edge technologies (*Herodotou et al., 2024*; *Kaplan Mintz et al., 2023*). By making the process more enjoyable (i.e. gamification), we can potentially gather even more biological data from the public (*Bowser et al., 2013*; *Ponti et al., 2015*). Yet, the collection process of Community-sourced data is usually not well-designed (e.g. spatially biased 'presence-only' data) (*Feldman et al., 2021*; *Steen et al., 2019*) and its interpretation is challenging without proper statistical modelling. Thus, although much effort has

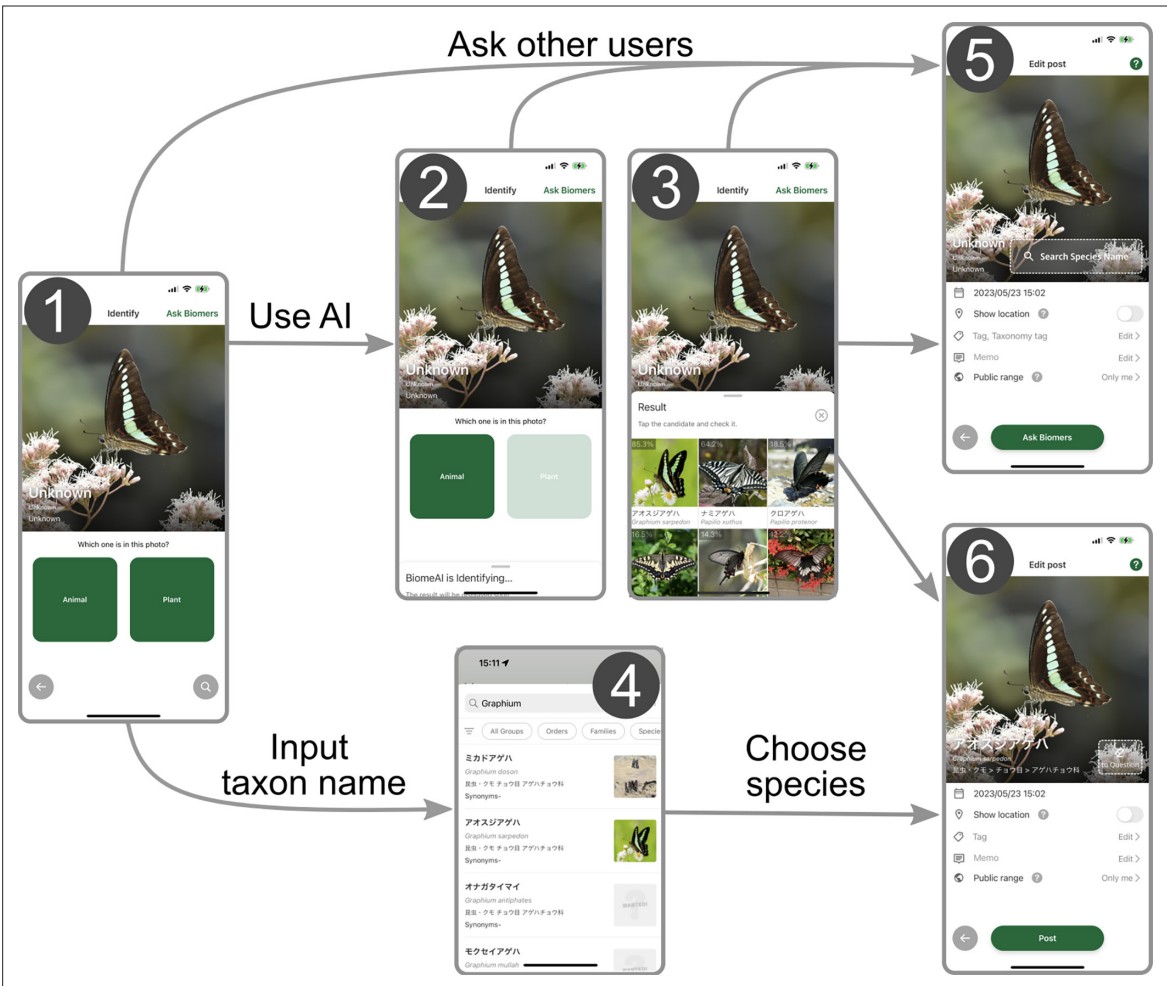

**Figure 1.** Workflow of submitting records to *Biome*. (1) Users can upload images that were taken by the smartphone camera or import existing images from the storage, including those imported from external devices. (2) Users select whether the image is about animals or plants to activate the species identification artificial intelligence (AI). (3) The AI analyses the image and its metadata to generate a candidate species list. (4) Alternatively, users can input the taxon name manually and obtain a list of candidate species. To submit the occurrence record, users can either (5) seek identification assistance from other users through the 'ask Biomers' feature, or (6) identify the species from the list. To the records, users can add memos and tags indicating phenology, life stage, sex, and whether the individual is wild or captive.

been invested in developing effective monitoring and modelling methods for biodiversity assessment, current approaches can be further improved by incorporating (i) more enjoyable community-based survey platforms using mobile applications and (ii) employing an advanced statistical modelling framework in estimating species distribution.

To fuel communities' engagement in biodiversity surveys and environmental education, we launched the mobile application *Biome* in 2019 in Japan (*Fujiki and Tatsuno, 2021*). For supporting species identification, *Biome* implements artificial intelligence (AI) algorithms that generate lists of potential species and enable users to seek help/suggestions from others for species identification (*Figure 1*) as in other applications such as iNaturalist and eBird. The unique feature of *Biome* is gamification which offers enjoyable experiences and facilitates communication among users (*Fujiki and Tatsuno, 2021*; *Koide et al., 2023*). For example, users can earn 'points' by contributing in various ways such as submitting records and suggesting species identifications to others, and their levels are determined based on the total points earned. The inclusion of networking and gamification elements can attract a wider user base, including those who may not typically engage in community science (*Bowser et al., 2013*; *Groom et al., 2021*). Consequently, *Biome* has accumulated data rapidly. Since its launch, 6 million records have been collected through the app (by 17 October 2023). This is more than four times greater than the number of records accumulated by the Global Biodiversity Information Facility (GBIF) from any data sources including iNaturalist and eBird during the same period in Japan (ca. 1.3 million). The data gathered through the app has been used for conservation planning and facilitating companies' financial disclosures by supplying and analysing species occurrence records.

Species distribution models (SDMs) are effective statistical tools for assessing biodiversity at specific sites while accounting for biases in survey efforts. SDMs use species occurrence records and environmental conditions to estimate the potential geographic ranges and suitable habitats for species (*Booth et al., 2014*; *Box, 1981*; *Elith et al., 2011*; *Hutchinson, 1957*; *Phillips et al., 2006*). These models play a crucial role in conservation and restoration planning by helping predict how changes in land use and climate impact species distributions (*Kindt, 2023*; *Porfirio et al., 2014*; *Urban et al., 2016*). While species presence/absence data—which needs extensive surveys by experts—is limited, presence-only data—which can be obtained from communities' observations—is much more available. MaxEnt (*Phillips et al., 2006*; *Phillips and Dudík, 2008*) is one of the most popular SDM methods due to its computational efficiency and estimation accuracy (*Valavi et al., 2022*). It can estimate species distribution from presence-only data by maximising the entropy of the probability distribution while satisfying constraints based on the available information (*Elith et al., 2011*; *Phillips and Dudík, 2008*). Since MaxEnt only requires occurrence records, it is well-suited for empowering community-based observations to predict species distributions. Also, while community-sourced data often suffer from spatially biased sampling efforts (i.e. sampling tends to concentrate in densely populated or touristic areas; *Kendal et al., 2020*; *Reddy and Dávalos, 2003*), SDMs such as MaxEnt can account for such spatial biases by considering the spatial distribution of sampling efforts when selecting pseudo-absence (background) locations (*Milanesi et al., 2020*; *Phillips et al., 2009*). When sampling efforts are adequately controlled, adding community-sourced data improves the accuracy of SDMs (*Johnston et al., 2018*; *Robinson et al., 2020*; *Steen et al., 2019*). This implies that SDMs may be substantially improved by utilising rapidly accumulating *Biome*'s species occurrence records if we adequately control the sampling efforts.

Here, we show the quality of community-based data gathered through the smartphone app *Biome* and how the data improves the prediction accuracy of species distribution. First, we assess the quality of occurrence records by investigating the fractions of non-wild and misidentified records. Second, we built SDMs based on two types of data: (i) traditional survey data (e.g. forest inventory census, museum specimens, and records extracted from published researches) only and (ii) a mixture of traditional survey and *Biome* data. We then compare the performance of the two SDMs. We modelled the distributions of 132 terrestrial animals and seed plants in the Japanese archipelago which covers subtropical to boreal areas. We finally discuss how our SDMs relying on community-sourced data may contribute to meeting the goals of GBF.

# Results

## The amount and quality of *Biome* data

By 7 July 2023, *Biome* had accumulated 5,275,457 occurrence records of 40,957 species across the Japanese archipelago (*Figure 2A*). The amount of occurrence records submitted to *Biome* has increased across the years (*Figure 2B*). On average, in 2022, users submitted 5407 records per day. The distribution of data along environmental gradients somewhat differs between *Biome* and Traditional survey data. To elucidate this distinction, we employed principal component (PC) analysis to summarise all environmental variables. The two datasets demonstrated divergent distribution patterns along PC1 (*Figure 2C*). This component, accounting for 6.1% of the total variation, is primarily influenced by land use, topography, and climate (*Supplementary file 1*). Among the environmental variables, a notable contrast between the datasets was observed in relation to the natural-urban gradient. The *Biome* data exhibited a relatively uniform distribution encompassing the entire gradient, while Traditional survey data was substantially biased towards natural areas (*Figure 2C*). The majority of records are attributed to insects (31.2%) and seed plants (41.8%), which are relatively accessible and can be easily photographed using smartphones (*Figure 2D*).

Out of all the records submitted to *Biome*, a total of 2,373,303 records (45.0%) successfully passed through the automatic filtering process. This dataset, referred to as the *Biome* data, is utilised for subsequent investigations. The quality of *Biome* data varied across taxa and the rarities of species (*Table 1*). The fraction of the records of wild individuals exceeded 97% in insects and birds, while it was lower than 90% in molluscs, seed plants, mammals and fishes. Among the records of wild individuals, at the species level, identification accuracy was higher than 95% in birds, reptiles, mammals, and amphibians but less than 90% in insects, fishes, and seed plants. At the genus level, identification accuracy was higher than 90% in all taxa except for insects. In the case of fishes and seed plants, identifications became 5–6% more accurate at the genus level compared to the species level. The family was correctly identified in more than 94% of records in all taxa examined. Common species had higher identification accuracy than rare species (average value, 95% vs. 87%). This tendency was prominent in insects and seed plants, but less in the other taxa. These results suggest that identifying rare species in taxonomically diverse taxa (i.e. seed plants and insects) is a challenging task.

## The performance of SDMs

SDMs using *Biome* + Traditional data, including *Biome* data at 50%, were more accurate than those modelled only using Traditional survey data when the two datasets have the same amount of occurrence records (*Figure 3*). Our analysis revealed that although the intercept of the Boyce index (BI, model accuracy metric that ranges between –1 and 1) did not differ between the two datasets (generalised linear mixed model, see 'Methods': $\beta = 0.02 \pm 0.03$, $t = 0.60$, p=0.55), *Biome* + Traditional data consistently led to a more rapid increase in SDM accuracy as the amount of data increased, compared to models solely relying on Traditional survey data ($\beta = 0.02 \pm 0.01$, $t = 3.72$, p<0.001).

When compared to SDMs using Traditional survey data, those using Biome + Traditional data achieved a high level of accuracy with a much smaller amount of data. For instance, BI, which ranges from –1 to 1, exceeds 0.9 with 294 ± 471 records (mean ± SD across all species) in the Biome + Traditional data, whereas the Traditional survey data requires 2129 ± 4157 records to achieve the same accuracy. This was also true in endangered species (included in Japanese national or prefectural red lists); although 2336 ± 3718 Traditional survey records were required to exceed 0.9 of BI, only 338 ± 571 were required for Biome + Traditional data.

Because we controlled the proportion of *Biome* data within the Biome + Traditional data as 50%, the amount of records of the Biome + Traditional data is often limited. In cases where a species had less *Biome* data compared to Traditional survey data, the total amount of records of Biome + Traditional data ends up being smaller than that of Traditional survey data alone. Therefore, the two datasets did not differ in the best model performances in each species (BIs of Biome + Traditional data: 0.81 ± 0.20; Traditional survey data: 0.83 ±0.20).

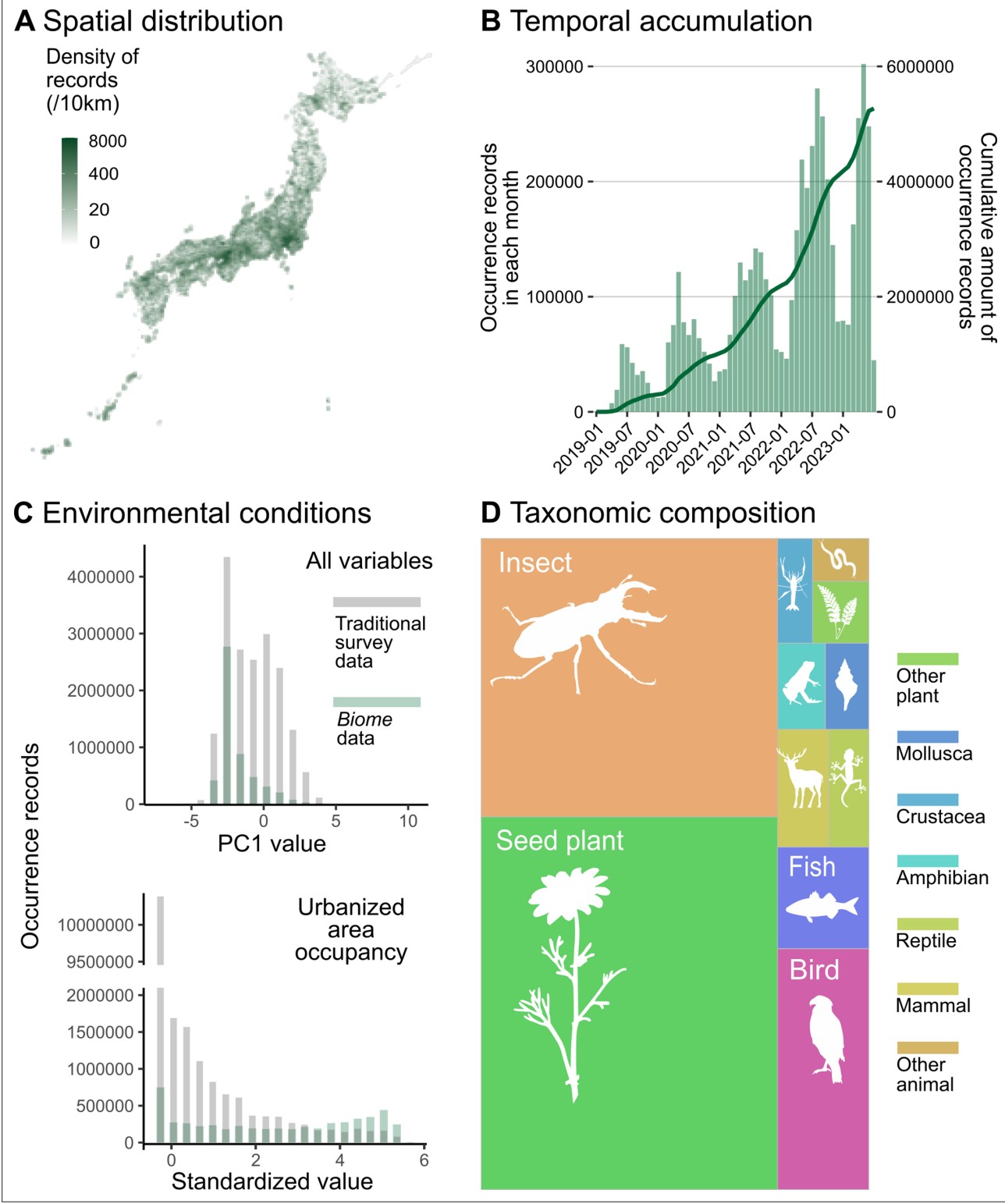

**Figure 2.** Description of data accumulated by *Biome*. Data distributions are shown based on all records submitted to *Biome* by 7 July 2023 (N = 5,275,457). (**A**) Spatial distribution of records across Japan. (**B**) Accumulation of records through time. The barplot represents the number of records each month and the line shows the cumulative amount of records. (**C**) Distributions of records along with PC1 of all environmental variables and standardised area occupancy of urban-type land uses. Grey and green represent distributions of Traditional and *Biome* data, respectively. (**D**) Taxonomic composition of records is shown as the area sizes. 'Other plant' consists of non-seed terrestrial plants; 'insects' include Arachnids and Insects; 'arthropods' cover any Arthropod not included in insects; 'other animals' covers all invertebrates not included in the taxa above.

**Table 1.** Data quality of *Biome*.

The fraction of records documenting wild individuals, and identification accuracy at species, genus, and family levels among the records documenting wild individuals are shown. Species were identified only for records documenting wild individuals.

| Species group | Species rarity | N | Wild/total (%) | Species correct/wild (%) | Genus correct/wild (%) | Family correct/wild (%) |
|---|---|---|---|---|---|---|
| Total | Total | 1420 | 81.6 | 91 | 93.6 | 96.9 |
| Seed plant | Total | 290 | 86.2 | 89.6 | 94.4 | 97.2 |
| Mollusca | Total | 140 | 87.9 | 90.2 | 91.1 | 96.7 |
| Insect | Total | 290 | 100 | 83.4 | 86.9 | 94.1 |
| Fish | Total | 140 | 73.6 | 87.4 | 93.2 | 96.1 |
| Amphibian | Total | 140 | 93.6 | 96.2 | 96.2 | 98.5 |
| Reptile | Total | 140 | 91.4 | 97.7 | 100 | 100 |
| Bird | Total | 140 | 98.6 | 98.6 | 99.3 | 99.3 |
| Mammal | Total | 140 | 80.7 | 95.6 | 95.6 | 96.5 |
| Total | Rare | 710 | 88.7 | 87 | 91 | 95.6 |
| Total | Common | 710 | 91 | 95 | 96.3 | 98.3 |
| Seed plant | Rare | 145 | 80.7 | 82.9 | 91.5 | 94.9 |
| Seed plant | Common | 145 | 91.7 | 95.5 | 97 | 99.2 |
| Mollusca | Rare | 70 | 82.9 | 86.2 | 87.9 | 96.6 |
| Mollusca | Common | 70 | 92.9 | 93.8 | 93.8 | 96.9 |
| Insect | Rare | 145 | 100 | 75.2 | 80 | 91.7 |
| Insect | Common | 145 | 100 | 91.7 | 93.8 | 96.6 |
| Fish | Rare | 70 | 74.3 | 88.5 | 94.2 | 94.2 |
| Fish | Common | 70 | 72.9 | 86.3 | 92.2 | 98 |
| Amphibian | Rare | 70 | 95.7 | 95.5 | 95.5 | 98.5 |
| Amphibian | Common | 70 | 91.4 | 96.9 | 96.9 | 98.4 |
| Reptile | Rare | 70 | 94.3 | 95.5 | 100 | 100 |
| Reptile | Common | 70 | 88.6 | 100 | 100 | 100 |
| Bird | Rare | 70 | 97.1 | 98.5 | 100 | 100 |
| Bird | Common | 70 | 100 | 98.6 | 98.6 | 98.6 |
| Mammal | Rare | 70 | 81.4 | 91.2 | 91.2 | 93 |
| Mammal | Common | 70 | 80 | 100 | 100 | 100 |

## Discussion

### *Biome*: The amount and quality of submitted data

Since its launch in 2019, the app *Biome* has accumulated species occurrence data rapidly (*Figure 2*). Despite our concerted efforts to engage non-expert users through gamification features, it is important to acknowledge that an excessive influx of non-expert users could potentially compromise the quality of the collected data. This could manifest in misidentifications or incomplete documentation, such as failing to appropriately label non-wild individuals. We thus have developed algorithms to exclude such suspicious records based on the features of records and users' behaviour on the app. The implementation of automatic data filtering techniques is expected to enhance the quality of the data, although further refinement is necessary. Notably, for insects and birds, which encompass numerous

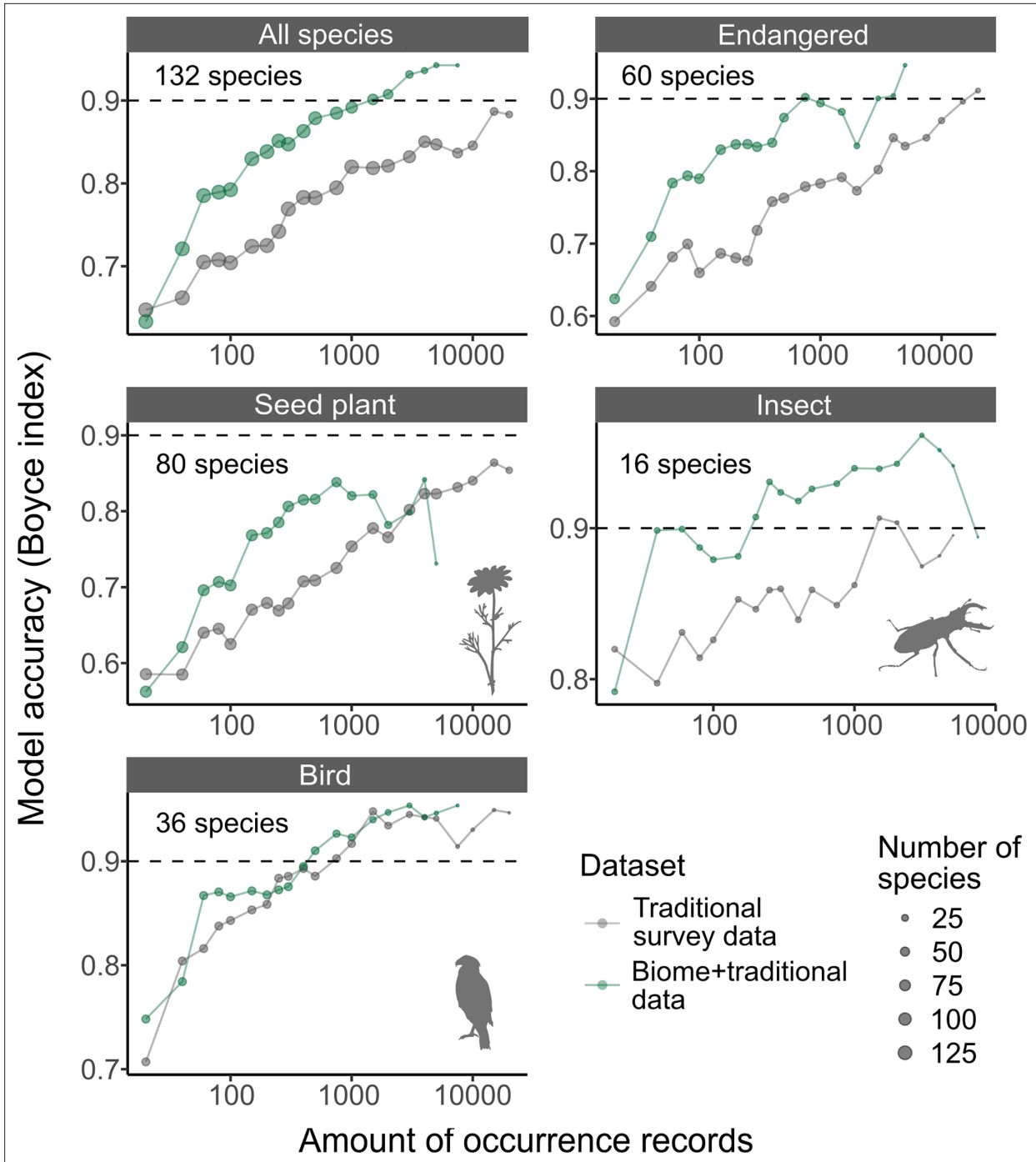

**Figure 3.** The accuracy of species distribution models. Accuracy of species distribution models (SDMs) using Traditional survey data (grey dots and lines) and Biome + Traditional data (i.e. 50% of *Biome* data: green). Each SDM was performed with a specific dataset, species, and the amount of records. For each species and amount of records, we computed the average model accuracy (Boyce index) from three replicated runs. Subsequently, we calculated the median model accuracy across species for each amount of records. These medians were then illustrated for each taxon in the strip of each respective panel. The 'Endangered' category includes species that are listed as endangered on Japan's national or prefectural red lists.

The online version of this article includes the following figure supplement(s) for figure 3:

**Figure supplement 1.** Accuracy of species distribution models (SDMs) using Traditional survey data (grey dots and lines) and Biome + Traditional data (i.e. 50% of *Biome* data: green), evaluated against test data only consisting of Traditional survey data.

species that can be kept in captivity, the majority of records that underwent filtering procedures were restricted to observations of wild individuals. Yet, the fraction of non-wild individuals is high in several taxa such as fishes and seed plants. In response, we have updated the posting flow in the app to prompt users to differentiate between non-wild and wild individuals. Further analysis is warranted to evaluate the impact of this update on data quality.

Once we could exclude non-wild individuals, species identification accuracy exceeded 95% in taxa with moderate species diversity (amphibians, reptiles, birds, and mammals). In seed plants, *Biome*'s species identification accuracy was 90%, which is higher than the accuracy of auto-suggest identification by commonly used apps for plants (69%, PlantNet, PlantSnap, LeafSnap, iNaturalist, and Google Lens; *Hart et al., 2023*). During the invasive plants survey in the United States, the reports by non-professional volunteers were 72% correct (*Crall et al., 2011*). The higher accuracy of species identification in *Biome* data can be attributed to two key factors. Firstly, the vigilant oversight of the user community through the 'suggest identification' feature plays a crucial role. *Biome* encourages users to participate in suggesting identifications by offering 'points' as rewards for their contributions. Secondly, the species identification AI algorithm leverages past occurrence data from nearby areas, resulting in increasingly accurate automatic identifications as the data accumulates. Given these, as a community science app, the data quality of *Biome* is decent. Yet, rare species generally showed lower identification accuracy, which would require identification by experts and further improvement of species identification AI algorithm.

## Species distribution modelling

The inclusion of *Biome* data resulted in improved accuracy of SDMs (*Figure 3*). The most accurate model predictions were obtained when the training data consisted of 50–70% *Biome* data (Appendix 1), highlighting the necessity of incorporating both traditional surveys and citizen observations for a comprehensive understanding of species distributions (*Miller et al., 2019*; *Pacifici et al., 2017*; *Robinson et al., 2020*).

The improvement can be attributed to introducing data with different biases compared to the Traditional survey data. Indeed, when controlling for the number of occurrence records, the model performance was higher in the Biome + Traditional data compared to the Traditional survey data. The variation in performance can be attributed to the distribution of data in relation to environmental conditions. Traditional survey data exhibits a strong bias towards natural areas, whereas *Biome* data is well balanced across the natural-urban habitat gradients (*Figure 2C*). Therefore, incorporating *Biome* data could significantly enhance modelling accuracy in urban and suburban landscapes, which are typically underrepresented in traditional survey data. As pseudo-absences are selected based on search effort, our models utilise numerous pseudo-absences from these areas. Consequently, this might lead to better estimation of species absence in such areas, not just presence, resulting in an overall increase in model accuracy across a wider range of species. A balanced distribution, along with the natural-urban gradient, is noteworthy because community science data is typically biased towards human population centres (*Kendal et al., 2020*; *Reddy and Dávalos, 2003*). This could be influenced by the distribution of users' residencies, although we do not have specific information about the users' locations. The app has collaborated with numerous local governments across Japan, including 9 prefectures and 29 local municipalities such as cities and towns. Through these collaborations, the user base may be widely dispersed, enriching the geographical coverage of *Biome* data.

The *Biome* data also can improve SDM accuracy by simply increasing the overall amount of data. Essentially, SDM accuracy is enhanced with an increased amount of data (*Figure 3*; *Erickson and Smith, 2023*; *Stockwell and Peterson, 2002*). In our analysis, we maintained a fixed proportion of 50% for *Biome* data within the Biome + Traditional dataset, which in turn restricted the amount of available Biome + Traditional data. However, our preliminary analysis (Appendix 1) demonstrates that the enhancement of SDM accuracy occurs across a range of proportion variations for *Biome* data blending. This implies that the proportion of *Biome* data does not necessarily need to be controlled. Therefore, in practical application scenarios, the incorporation of *Biome* data predominantly serves to augment the overall volume of training data.

The impact of community-sourced data on SDMs has primarily been investigated using birds, with a limited focus on plants (*Feldman et al., 2021*). In our investigation, we observed that incorporating *Biome* data improved SDM accuracy for seed plants and insects, while the impact on birds

remained unclear (*Figure 3*). This ambiguity is likely because community-sourced data from platforms such as eBird are already incorporated in Traditional data through GBIF. In comparison to other taxonomic groups, our results indicate that seed plants exhibited lower model accuracy when evaluated against both Biome + Traditional survey data (*Figure 3*) and Traditional survey data alone (*Figure 3—figure supplement 1*). The variation in model accuracy among taxonomic groups may be attributed to data quality issues in both *Biome* and Traditional survey data. For instance, in *Biome* data, while the fractions of wild individuals were high in birds and insects, it was lower for seed plants (*Table 1*). Compared with other taxa, distinguishing between wild and non-wild individuals can be particularly difficult in plants when they are planted outside. In addition, identifying plant species may be challenging in certain taxa, primarily due to the absence of key identification traits on leaves and stems. This becomes especially problematic when flowers are not present. These difficulties could potentially impact the quality of Traditional data as well. Although few studies have simultaneously assessed the quality of community-sourced data and its impact on SDMs across different taxa, it is important to recognise that data quality can vary among taxa.

Importantly, SDMs for endangered species, which often suffer from data deficit (*Erickson and Smith, 2023*; *Wisz et al., 2008*), became accurate in a much fewer amount of records by blending *Biome* data (*Figure 3*). Specifically, a threshold of >0.9 BI could be reached with only around 300 records when using *Biome* data, whereas over six times of data is required when using Traditional survey data only. This finding highlights the importance of community-sourced data not only for monitoring the dynamics of endangered species (*Chandler et al., 2017*; *Zapponi et al., 2017*) but also for modelling purposes. Considering the rapid accumulation of *Biome* data, *Biome* data would make a significant contribution to the more effective distribution modelling of endangered species.

## Limitations of this study

In assessing data quality, reidentification was impossible for records that did not photograph key traits for species identification. To address this limitation, further app improvements can include allowing users to submit multiple images. Encouraging users to document various body parts of organisms through multiple images would make capturing key identification traits much easier. This will make reidentification easier and possibly improve automatic species identification accuracy.

Given the absence of a comprehensive, environmentally unbiased occurrence dataset spanning a wide range of taxa, we assessed SDM accuracy not relying on an independent test dataset. In this evaluation, the test data was meticulously crafted to include 25% *Biome* data, serving as an intermediary proportion between Biome + Traditional (50%) and Traditional survey data (0%). By leveraging the distinct distribution patterns of *Biome* and Traditional survey data along environmental variables (*Figure 2C*), the test data would better encapsulate the actual species distribution compared to datasets composed solely of either *Biome* or Traditional survey data. It is noteworthy that, even when the test data exclusively consisted of Traditional survey data (i.e. unfavourable conditions for Biome + Traditional data SDMs), the accuracy of SDMs derived from Biome + Traditional and Traditional survey data did not differ (*Figure 3—figure supplement 1*). This result further supports our conclusions that *Biome* provides valuable data for SDM in terms of the amount and quality, and that blending *Biome* data improves SDM accuracy.

We evaluated SDMs based on spatial transferability using the central Japan region, which encompasses a range of environmental conditions. However, the evaluation results may not necessarily indicate transferability across the entire Japanese archipelago. Instead, in the near future, we anticipate that we can evaluate SDM accuracy using temporal transferability. The rapid accumulation of *Biome* data will allow us to evaluate the temporal transferability using the occurrence dataset from different time periods, and thus enable assessing their performance in much wider regions. In addition, limited data availability for certain taxa hindered the assessment in those taxa (e.g. molluscs, amphibians, reptiles, and mammals), but *Biome* would be a platform to overcome the data limitation for many taxa.

Finally, our SDMs do not directly indicate the species' presence probability. The output from presence-only SDMs usually deviates from the probability of presence when species prevalence (i.e. the proportion of area where the species occupied, requiring presence/absence data throughout the area) is unavailable (*Elith et al., 2011*; *Ward et al., 2009*). Due to the unavailability of absence data, SDM outputs in this work are indirect measures of species presence and thus are not directly

comparable across different species. Nonetheless, they are comparable within a species, providing useful information for understanding species distributions.

## Future directions

By blending data from traditional surveys and communities, we improved the accuracy of species distribution estimates. This enhanced estimation lays the groundwork for more precise subsequent analyses. For instance, estimated distributions will be useful in selecting new protected areas or areas with Other Effective area-based Conservation Measures (OECMs): allowing a wider range of land use as long as biodiversity and ecosystem services are sustained/improved. Using estimated distributions of each species, hotspots of species or evolutionary diverse taxa can be inferred. Such sites will be good candidates for protected areas (*Jones et al., 2016*) or OECMs (*Shiono et al., 2021*). Further, estimated distributions can be used as input for spatial conservation prioritisation tools (e.g. Marxanl *Ball et al., 2009*).

In our experience, stakeholders—including corporate social responsibility managers and conservation practitioners—often seek the list of species potentially inhabiting their locations. Due to the uncertainty of SDMs and their thresholding into presence/absence, on-site surveys remain essential for assessing biodiversity status. Yet, SDMs can make such surveys cost-effective by screening important locations for on-site assessment (e.g. Locate phase in TNFD framework) and narrowing down the target species for surveying. Improved estimation through SDMs can mitigate the risks associated with their use in society and enable more informed decision-making for conservation efforts.

The rapid accumulation of data from diverse locations holds the potential to unveil valuable ecological patterns. The accumulated data enables early detection capabilities for range expansions of invasive species (Sakai et al., in preparation). For instance, *Biome* data has hinted at potential range expansions in several insect species, including butterflies, dragonflies, and stink bugs, as well as changes in wintering areas for birds (*Biome Inc, 2023*). Given the diverse taxonomic coverage of *Biome* data (*Figure 2D*), detecting phenological changes across various taxa may be possible. This, in turn, is useful in uncovering phenological mismatches exacerbated by climate change, which can significantly change the dynamics of interacting species (*Renner and Zohner, 2018*; *Visser and Gienapp, 2019*). Moreover, *Biome* data is well-suited for assessing the effects of urbanisation on ecosystems since it comprehensively spans both urban and natural habitats (*Figure 2C*). The benefit of rapidly accumulating data, combined with recent advancements in machine learning methods, opens up opportunities for conducting time-series analyses. Community science data has rarely been used for time-series population analysis due to its notable spatiotemporal bias in sampling efforts (*Feldman et al., 2021*; *Zhang et al., 2021*). However, the two-step machine learning approach, as demonstrated by Fink and colleagues in estimating bird population trends using eBird data (*Fink et al., 2023*), sets a precedent. In the future, *Biome* data may facilitate the inference of population dynamics for multiple taxa. This will enable various time-series analyses to unveil ecosystem stability and interaction strength, which holds potential for forecasting ecosystem dynamics (*Laubmeier et al., 2020*; *Pennekamp et al., 2019*; *Ushio et al., 2018*).

For financial disclosures, companies will assess how their activities rely on ecosystem services and their opportunities for protecting/recovering nature (*TNFD, 2023*). By incorporating taxon-specific ecosystem services, multifaceted ecosystem services can be preliminarily screened (*Kass et al., 2024*). For example, based on estimated distributions of bumblebees or insectivorous animals, the functioning of pollination services or pest regulation services might be inferred. Using counts of 'likes' or records from *Biome* data, the charismatic species can be determined. By identifying places with a high estimated richness of charismatic species, potential areas for ecotourism can be screened. Because SDMs allow us to simulate the impacts of changes in landuse and climate (*Porfirio et al., 2014*; *Urban et al., 2016*), we will be able to forecast how those changes may influence local biodiversity and/or ecosystem functioning. Hence, estimated distributions provide the basis of nature-related financial disclosures.

Our platform facilitates collaboration among diverse stakeholders, including local communities, landowners, and employees from both private companies and government agencies. Engaging a broader spectrum of stakeholders is crucial for effective biodiversity assessment, nature management planning, and nature-related financial disclosures: this inclusivity allows for the incorporation of traditional knowledge into planning processes, mitigates conflicts among stakeholders, and ultimately

supports more seamless and informed decision-making (*Chan et al., 2021*; *Keough and Blahna, 2006*; *Linsley et al., 2023*; *Roy et al., 2023*; *TNFD, 2023*). Supporting natural experiences for a wide range of people is also expected to contribute to changing people's minds towards nature. By experiencing nature, people become familiar with it and subsequently make pro-nature decisions (*Soga and Gaston, 2023*). We believe that community science can significantly contribute to KM-GBF and create a sustainable society by fostering nature-positive awareness in society and providing data tools that enable effective action.

## Methods

### Key resources table

| Reagent type (species) or resource | Designation | Source or reference | Identifiers | Additional information |
|---|---|---|---|---|
| Software, algorithm | R 4.1.3; MaxEnt (using ENMeval 2.0 package on R) | R 4.1.3 (*R Core Team, 2021*); MaxEnt (*Phillips et al., 2006*; *Phillips and Dudík, 2008*); ENMeval 2.0 package (*Kass et al., 2021*) | | |
| Other | Species occurrence data | Biome app, GBIF and others (see 'Methods') | For DOIs of GBIF data, see *Supplementary file 2* | For details, see section 'Occurrence data' |

### Occurrence record accumulation through the mobile app *Biome*

In April 2019, a free smartphone app called *Biome* was launched for the Japanese markets. The app has been downloaded 839,844 times by 13 September 2023. The app allows users to collect data on the distribution of plants and animals using their mobile devices. Users can post photographs of the plants and animals they find, and the app automatically records the location and timestamp from EXIF data. If the EXIF data is unavailable, users can manually input the locality and timestamp.

To support species identification, the app provides users with two options. First, the app provides a list of candidate species based on the image and metadata (e.g. location and timestamp). *Biome* employs a synergistic approach that integrates image recognition technology and geospatial data to facilitate species identification. The image recognition algorithm, constructed upon convolutional neural networks, classifies species at higher taxonomic levels. Subsequently, these candidates are refined based on their frequency of recent occurrences in the geographical area. Consequently, as the correctly identified records accumulate for a given area, species identification AI will improve the accuracy. Second, users can seek help from other users. If a user selects the 'ask Biomers' button, their occurrence record is added to a waiting list that appears on the home screen. Other users can suggest possible identifications for the records, as in other records of which species was already identified.

Users can view and comment on other users' records. However, for conservation purposes, *Biome* automatically conceals the geolocations of endangered species that are listed on the Japanese national or prefectural red lists. This feature sets it apart from iNaturalist, where users must manually choose to hide the location of endangered species (*Koide et al., 2023*). The social networking function provides opportunities for communication among users, including non-experts (*Fujiki and Tatsuno, 2021*). Users earn 'points' through their contributions, including record submissions and identification suggestions to other users, and progress to higher levels based on their total points. The points awarded depend on the rarity, conservation status, and societal impact of the species submitted, meaning that users earn more points when submitting records of rare, endangered, or invasive species. The app occasionally offers 'Quests' events that provide users with an opportunity to earn additional points by submitting records from specific locations or of particular species, crucial for monitoring phenology. Through the variety of gamification features, we stimulate people to participate in biological surveys as a fun activity.

We obtained occurrence records submitted to *Biome* by 7 July 2023. The raw data collected through *Biome* contains invalid presence records which we defined in the present study as unclear images, documenting non-wild individuals and misidentifications, and images including some privacy issues. To improve data quality, we excluded records deemed to be invalid mainly based on location metadata and users' reactions to the record is as detailed below. This filtered *Biome* data is used in the subsequent investigations.

## Filtering suspicious occurrence record in *Biome* data

Occurrence records of non-wild individuals were eliminated as much as possible by using the information provided by users and location of records. *Biome* users sometimes report inappropriate records (e.g. unclear images and images from websites or books), and we excluded all of those reported records. All private records were excluded because they can harbour inappropriate and misidentified records not being screened by other users. We also excluded occurrence records that users had marked as non-wild individuals: users have an option to label their records as photographing bred or cultivated individuals, or specimens. Records from cultural centres (i.e. zoos, botanical gardens, museums, and aquariums) and large pet stores were removed as well. During the data correction process, we prioritise the suggestions provided by *certified users* (see below for the definition), regardless of the decisions made by the users who originally created the record. Furthermore, we excluded records that have not been posted by *certified users* or have not received identification suggestions from *certified users*.

*Certified users* are defined as users who achieved the higher accuracy of species identification (<15% of public occurrence records were suggested as misidentification by other users), submitted few inappropriate records (<0.5% of public records), and have created >20 public records. We also defined *specialist users*, a subset of *certified users* identified in each taxa (see *Figure 2* for the classification), who made a total of >30 records or identification suggestions with high identification accuracy (the fraction of suggested records is less than the average of *certified users* in the taxa). *Specialist users* are used in determining pseudo-absence for SDMs.

## Assessing the accuracy of records

We investigated the proportion of occurrence records within the *Biome* data that were suitable for SDMs. Since SDMs are influenced by invalid presence records, we assessed the quality of *Biome* data based on a total of 1420 records from rare and common species of seed plants, molluscs, insects (including Arachnid and Insecta), fishes, mammals, birds, reptiles, and amphibians (*Figure 4*). We defined rare species as those with less than or equal to 10 occurrences in *Biome* data, and common species as those with the highest 15% of records in each taxonomic category. In each of the seed plant and insect species which account for the majority of *Biome* data (*Figure 2D*), we randomly selected 145 records of each rare and common species. For the other taxonomic categories, we chose each of the 70 records from rare and common species.

Records were first screened whether they targeted organisms (images with no organisms were discarded) and contained wild individuals. To assess the accuracy of species identification, species in the records that documented wild individuals were manually reidentified by experts with taxonomic knowledge (*Figure 4*). These experts have professional backgrounds, serving as a technician at a prefectural research institute (fish), highly experienced field survey conductors (plants and insects, respectively), a post-doctoral researcher (amphibians and reptiles, and mammals, respectively), and a museum curator (molluscs) specialising in the focal taxa. Then, by comparing species identifications by the experts and on *Biome* data, the results were classified into two categories: (1) correct based on the image and locality—based on the image, identification was probably correct, and the image locality matches with habitat/range of the species; (2) misidentification—records were reidentified by experts if possible. We also examined whether the identification was correct at genus and family levels.

## Species distribution models

### Occurrence data

To evaluate the impact of *Biome* data on SDM prediction accuracy, we compiled two datasets: 'Traditional survey data' and 'Biome + Traditional data'. The Traditional survey data comprised records collected through conventional survey techniques (e.g. riverine census, forest inventory census, and museum specimens) primarily sourced from the National Census on River and Dam Environments (NCRE) and GBIF. In contrast, the Biome + Traditional data encompassed records submitted to *Biome* that passed filtering methods, in addition to the Traditional survey data. To control the relative proportion of *Biome* data, we constrained the fraction of *Biome* data within the Biome + Traditional data to 50% for each species. Our preliminary results showed that blending 50–70% of *Biome* data in training data improved prediction accuracy (Appendix 1). For traditional survey data, we downloaded occurrence records of relevant taxa from GBIF between 20 April 2023. To prevent significant differences

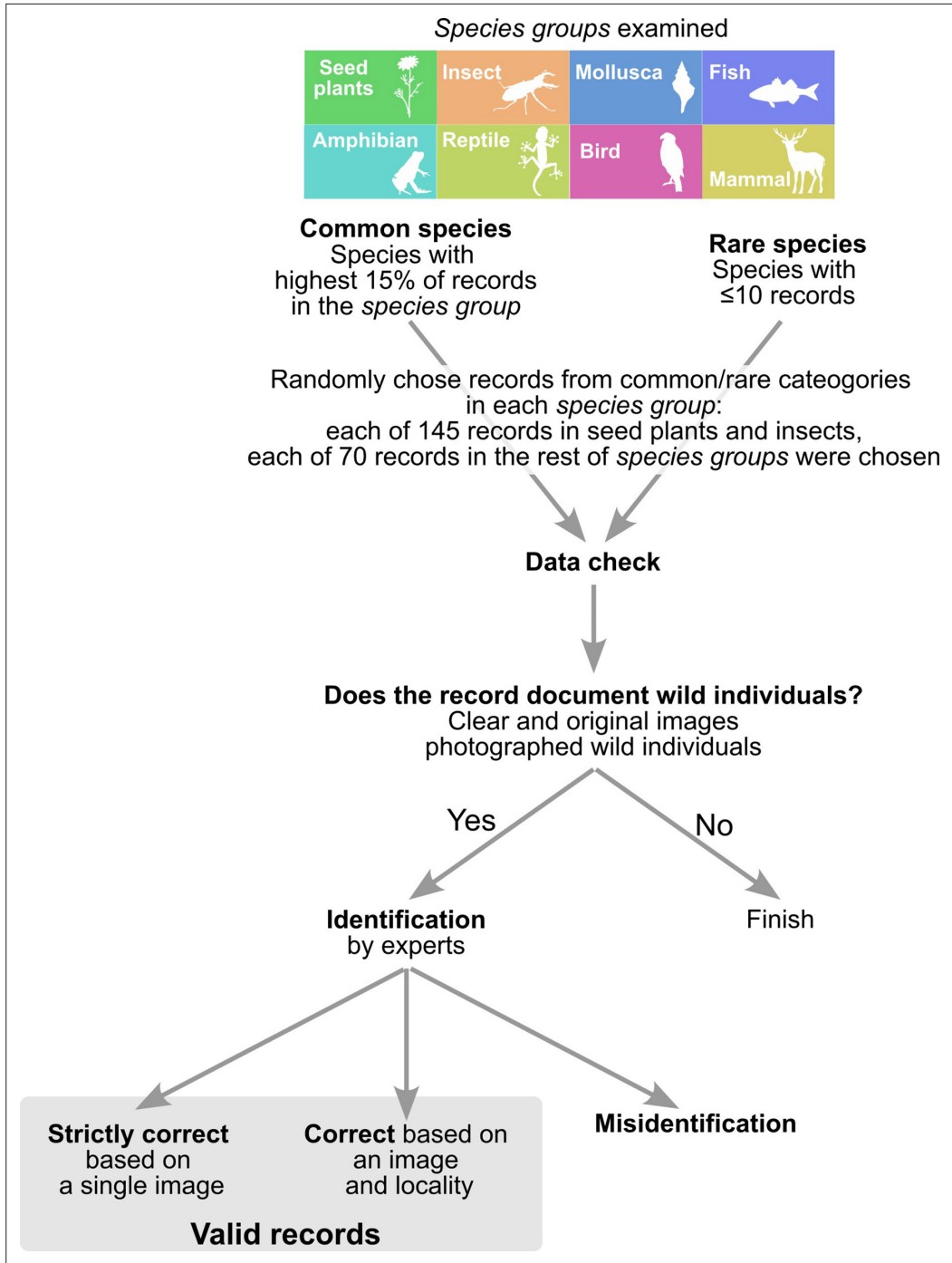

**Figure 4.** The workflow of checking accuracy of *Biome* data.

between the sampling periods of the GBIF records and environmental data, we used the GBIF sampled after 1970. The clean_coordinates function of the R package 'CoordinateCleaner' was used to remove records with erroneous coordinates such as records from country capitals and centroids, and biodiversity institutions. We obtained occurrence data from the large occurrence datasets such as the NCRE and Forest Ecosystem Diversity Basic Survey. For the areas or taxa where occurrences were scarce, we further compiled the literature with detailed locality information, such as local species inventories. The amount of occurrence records in the modelled species and species coverage of each dataset is summarised in *Table 2*. For the species analysed (S9 Table), traditional survey data contains

**Table 2.** List of species occurrence datasets used for constructing species distribution models (SDMs).
To compare *Biome* dataset with the other datasets, iNaturalist and eBird data based on community science were classified as 'Traditional survey' data.

| Original dataset | Occurrence records of modelled species | | Species coverage among modelled species | Survey method | Data group in SDM | Download date | Availability |
|---|---|---|---|---|---|---|---|
| | *N* | Occupancy | | | | | |
| *Biome* (filtering applied) | 201,114 | 8.6 | 132/132 | Citizen science through smartphone app | *Biome* | 7 July 2023 | https://biome.co.jp/ |
| National Census on River and Dam Environments (NCRE) | 1,413,541 | 60.2 | 126/132 | Traditional survey on freshwater and its adjacent ecosystems | Traditional survey | 10 January 2023 | http://www.nilim.go.jp/lab/fbg/ksnkankyo/ |
| Institute records registered at GBIF | 530,952 | 22.6 | 116/132 | Traditional survey and museum specimens | Traditional survey | 7 July 2023 | GBIF* |
| iNaturalist and eBird | 118,050 | 5 | 110/132 | Citizen science through smartphone app and web service | Traditional survey* | 7 July 2023 | GBIF* |
| Forest Ecosystem Diversity Basic Survey | 80,929 | 3.4 | 42/132 | Traditional survey on forest trees | Traditional survey | 30 March 2023 | http://forestbio.jp/ |
| Literature | 3293 | 0.1 | 130/132 | Traditional survey | Traditional survey | 31 March 2023 | Refs* |

*For the list of GBIF download doi and literature, see **Supplementary file 2**.

a negligible portion of community-sourced data (5.5%) because GBIF contains community-sourced data from iNaturalist and eBird.

## Predictor variables

Predictors encompass a range of environmental variables recognised to impact species distribution (**Table 3**): land use (**Newbold et al., 2015**), climate (bioclim variables; **Booth et al., 2014**), vegetation (**Abe, 2018**), lithology (**Ott, 2020**), and elevational range (**Udy et al., 2021**). Additionally, categorical

**Table 3.** Environmental data used for constructing species distribution models (SDMs).
Years indicate the data collection period. Usage in the SDM shows how the variables were converted before using in the species distribution modelling.

| Data | Variables | Year | Usage in the SDM | Available at |
|---|---|---|---|---|
| Land use | The area sizes of forests, rice fields, farms, wastelands, inland waters, beaches, ocean, golf courses, urbanised areas, and others | 2016 | Extracted six principal components (PCA) explained ≥ 80% of total variation. PCs were converted into linear, quadratic and hinge terms. | The Ministry of Land, Infrastructure, Transport and Tourism of Japan (MLIT) (https://nlftp.mlit.go.jp/ksj/gml/datalist/KsjTmplt-L03-a.html) |
| Forest type | Forest type (planted and natural) | 1998 | Converted into linear, quadratic, and hinge terms. | The Biodiversity Centre of Japan (http://gis.biodic.go.jp/webgis/index.html) |
| Climate | Monthly average, minimum and maximum temperature and precipitation | 11981–2010 | Transformed into 19 bioclimatic variables (**Booth et al., 2014**), then extracted three PCs explained ≥ 80% of total variation. Converted into linear, quadratic, and hinge terms. | MLIT (https://nlftp.mlit.go.jp/ksj/gml/datalist/KsjTmplt-G02-v3_0.html) |
| Elevation-al range | Differences between maximum and minimum elevation, and maximum slope | 1981 | Converted into linear, quadratic, and hinge terms. | MLIT (https://nlftp.mlit.go.jp/ksj/jpgis/datalist/KsjTmplt-G04-a.html) |
| Vegetation | The area sizes | 1998 | Transformed into 37 PCs of which total variation explained was more than 80%. Converted into linear, quadratic and hinge terms. | MOE (http://gis.biodic.go.jp/webgis/index.html) |
| Geology | The area sizes of limestone and serpentinite | 2022 | Converted into linear, quadratic and hinge terms | The Research Institute of Geology and Geoinformation (https://gbank.gsj.jp/seamless/use.html) |
| Geohistory | Blakiston's Line (**Dobson, 1994**; **Saitoh et al., 2015**), oceanic islands (**Wepfer et al., 2016**; **Yamasaki, 2017**) | | Categorical variables | |

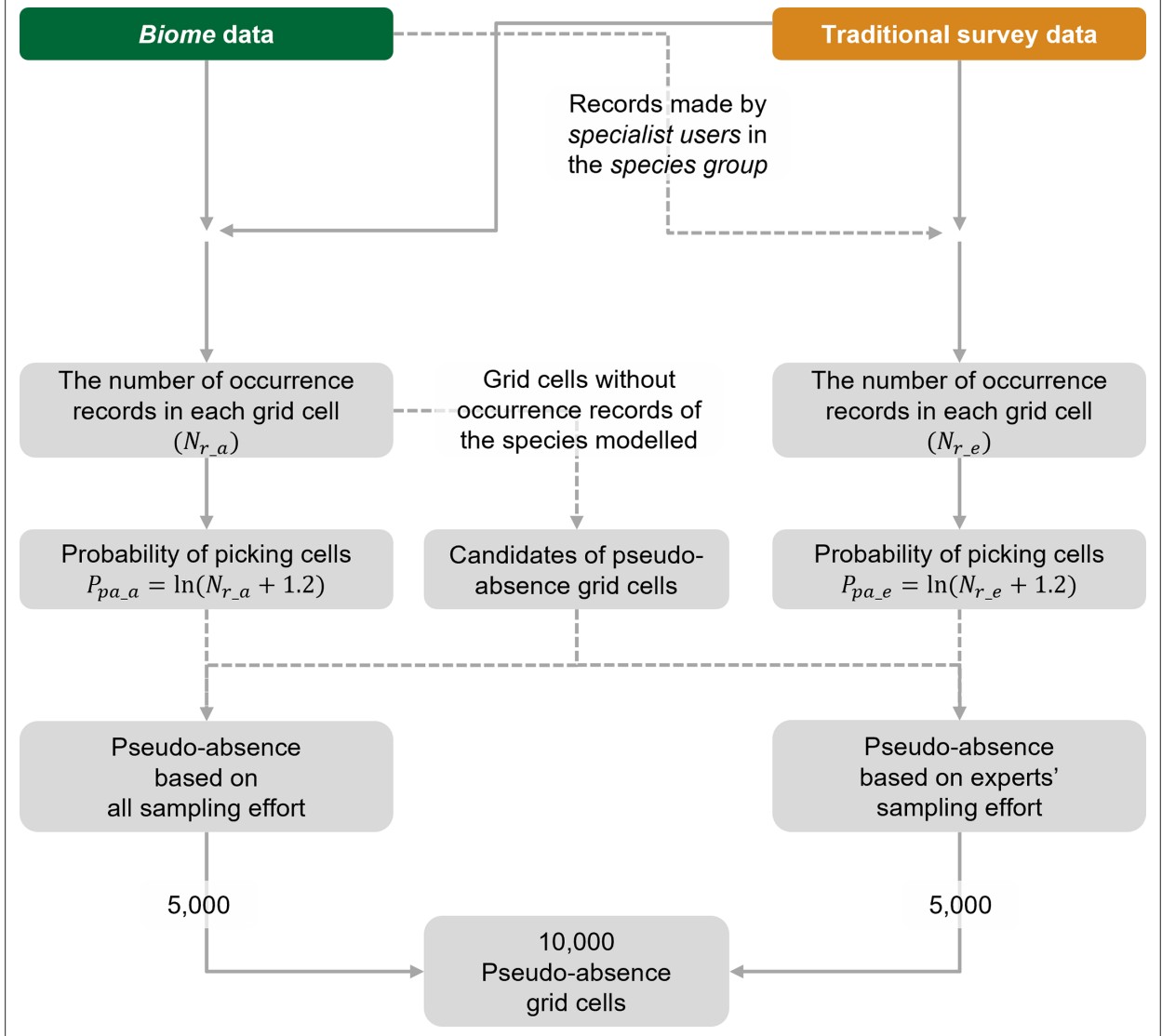

**Figure 5.** The workflow for selecting pseudo-absence (background) grid cells for species distribution models (SDMs) using the *Biome*-Traditional dataset. In this process, both *Biome* data and Traditional dataset are utilised to determine the suitable locations for pseudo-absence grid cells. However, when constructing SDMs using the Traditional dataset exclusively, *Biome* data is not involved in the selection of pseudo-absence points.

variables representing known biogeographic regions, reflecting geological history, were included. We applied Blakiston's Line—Tsugaru straits dividing the northern and main islands of Japan (i.e. Hokkaido and Honshu islands)— reflecting a significant historical migration barrier for mammals and birds (**Dobson, 1994**; **Saitoh et al., 2015**). Due to the distinct fauna (**Wepfer et al., 2016**; **Yamasaki, 2017**), we also specified oceanic islands (i.e. Ogasawara and Daito isles) which have never been connected with the Asiatic continents. Continuous environmental variables were transformed into linear, quadratic, and hinge feature classes to illustrate nonlinear associations between environments and species occurrence (**Phillips et al., 2017**). The regularisation multiplier was set at 2.5, falling within the established optimal range of 1.5–4 (**Elith et al., 2010**; **Moreno-Amat et al., 2015**).

## Pseudo-absence reflecting search effort

We considered sampling efforts when selecting a total of 10,000 pseudo-absence locations. To accommodate biases in sampling efforts, we assigned picking probabilities as an increasing function of the amount of occurrence records of all and relevant taxa at the grid cell (an index of sampling efforts) (**Milanesi et al., 2020**; **Phillips et al., 2009**). That is, grid cells with rich occurrence records of relevant

taxa are more likely to be chosen as pseudo-absences than cells with few records, as detailed below (see also *Figure 5*).

To generate pseudo-absence (i.e. background) data, we employed two approaches considering different sampling efforts. The first approach incorporated all observers and taxa, while the second approach focused on experts and relevant taxa (*Figure 4*). In both cases, pseudo-absences were selected from grid cells that lacked any occurrence records of the species being modelled. However, due to variations in sampling efforts across locations, it was important to address potential bias. To mitigate this bias, we adjusted the picking probability based on the number of occurrences of other species in each grid cell (*Milanesi et al., 2020*; *Phillips et al., 2009*).

In the first approach, we assumed that the users of *Biome* submit records of any taxon without specifically selecting species from particular taxa. The picking probability was simply determined by the total number of records from all taxa in the *Biome* data in every grid. In the second approach, we considered the expertise of observers (*Milanesi et al., 2020*) and the sampling effort for relevant taxa (*Phillips et al., 2009*). We also assumed that Traditional surveys targeted particular taxa. Under this approach, we selected records from *Biome* data contributed by *specialist users* and all records from the Traditional survey data. From this subset of data, we calculated the number of records for the taxa (e.g. seed plant, insect, and amphibian) to which the modelled species belonged. This information was then used to calculate the picking probability for each grid cell. To account for the variability in record counts among locations, we applied a logarithmic transformation to the number of records. We also added a value of 1.2 before taking logarithms to allow for the selection of pseudo-absences with low probabilities, particularly in locations with only one or no records of other species. Pseudo-absences were not chosen from the spatial block used as test data, but otherwise, there were no geographical restrictions on their selection.

Using the described approaches, we obtained a total of 10,000 pseudo-absences for our analyses. The amount of pseudo-absences follows the default setting of MaxEnt (*Elith et al., 2011*). For the models using Biome + Traditional dataset (also in *Biome*-blended dataset in Appendix 1), pseudo-absences were generated by merging each of the 5000 points identified through the two approaches. Meanwhile, for SDMs using the Traditional survey data only, we obtained 10,000 pseudo-absences by exclusively using the second approach without incorporating *Biome* data.

## Modelling

We modelled distributions of terrestrial seed plants and animals at a scale of 1 × 1 km grid cell, based on Traditional survey data and *Biome* + Traditional data. To model species distributions from presence-only data, several algorithms have been utilised, including generalised additive models, random forest, and neural networks (*Norberg et al., 2019*; *Valavi et al., 2022*). In our study, we opted for MaxEnt (*Phillips and Dudík, 2008*) due to its high estimation accuracy and relatively low computational burden (*Valavi et al., 2022*). We performed MaxEnt via ENMeval 2.0 package (*Kass et al., 2021*) on R 4.1.3 (*R Core Team, 2021*).

## Model evaluation

We evaluated the model by examining spatial transferability because we could not find occurrence data that are environmentally unbiased and independent from training data. To minimise spatial auto-correlation between training and test data, we set a spatial block for splitting data (*Araújo et al., 2019*; *Santini et al., 2021*). As the spatial block, we chose the central Japan region (latitude, 33.7°–37.7° N; longitude, 136.2°–137.6° E: *Figure 6*) which covers various environments—alpine to coastal lowlands, metropolis to highly intact areas.

To ensure a fair and balanced assessment of the accuracy of SDMs built from Traditional survey data (0% *Biome* data) and Biome + Traditional data (50% *Biome* data), we compiled a test dataset that embodies characteristics intermediate between these two datasets. This composite test dataset encompasses 25% *Biome* data and 75% Traditional data, effectively bridging the differences between the two original datasets and providing a comprehensive basis for evaluating SDM accuracy.

Due to the presence of invalid records, *Biome* records were used as test data only when multiple users recorded the same species within an identical 1 km grid cell. Although *Biome* data may include invalid records (i.e. non-wild individuals or misidentification), if multiple users recorded the same species at the same place, any one of the records from the place is likely to be valid. As we know

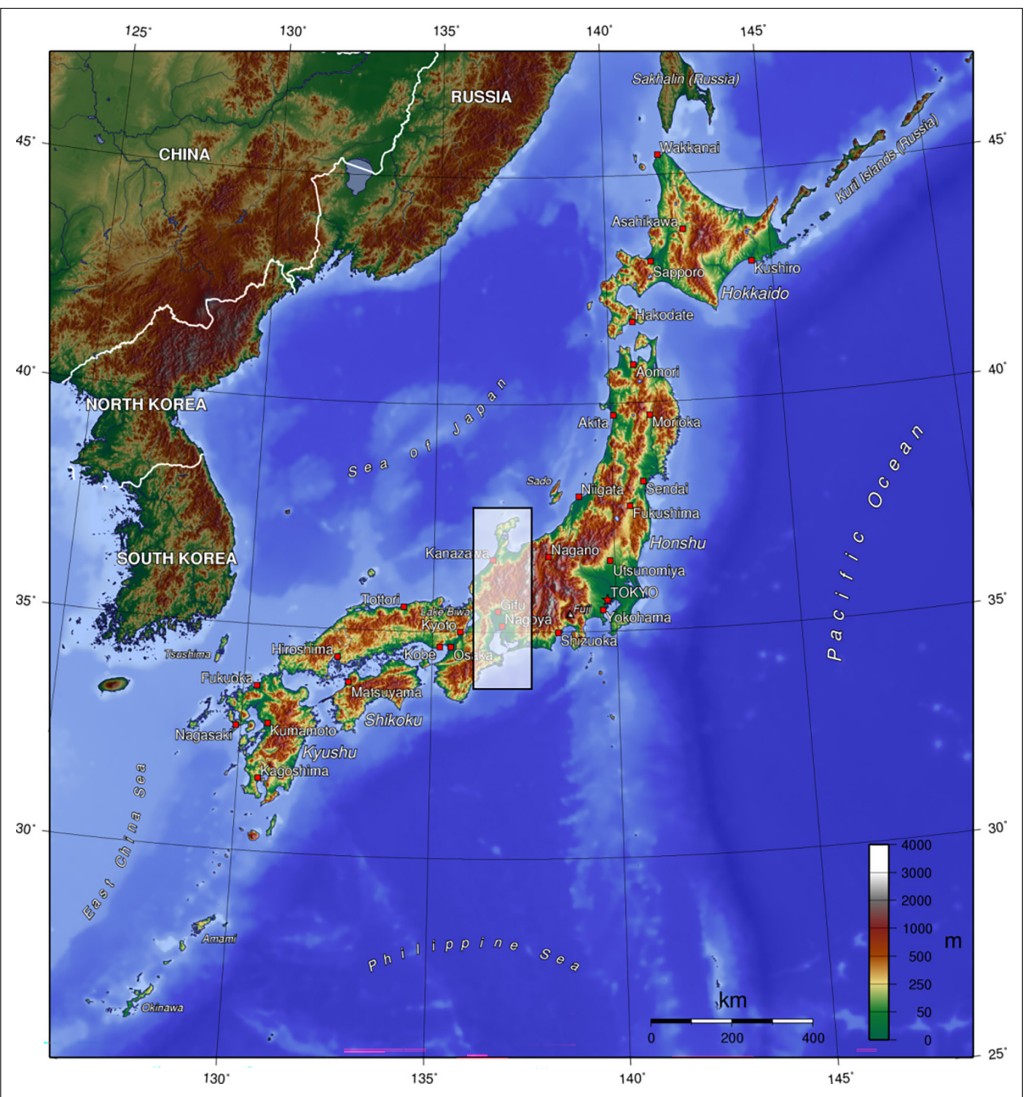

**Figure 6.** Japanese archipelago, coloured by altitude. Shaded area shows spatial block of test data. Retrieved from Wikipedia (2023, May 30), licensed under Creative Commons Attribution-ShareAlike 3.0 Unported (CC BY-SA 3.0).

the fraction of valid records within the *Biome* dataset in each taxon (see Results), we can calculate the probability of the true presence in a given location as follows, by assuming that records made by different users were independent:

$$p_{tp} = 1 - (1 - p_{valid})^{n_{users}}$$

The probability of valid records at a given taxon is shown as $p_{valid}$, and the number of users reported given species at the place is indicated as $n_{users}$. If $p_{tp}$ exceeds 99%, we deemed that the species occurred in the location.

To reduce spatial sampling bias, we downsampled a dataset within Traditional survey data, NCRE with massive records from freshwaters, to match the number of records from the remaining Traditional survey data. This procedure is applied to all test datasets in both the main analysis and preliminary analyses documented in *Figure 3—figure supplement 1* and Appendix 1.

BI was used to measure model performance because it was designed to evaluate presence-only SDMs (*Hirzel et al., 2006*). In short, BI measures the correlation between estimated habitat preference and the frequency of actual presence, and ranges from –1 to 1. A high BI indicates high SDM accuracy that presence data points tend to be located in grids with higher habitat suitability values. To reliably calculate BI, at least 50 occurrences should be needed in test data (*Hirzel et al., 2006*). Thus, we used 132 species that have more than 50 occurrences in test data for calculating BI (*Supplementary file 3*).

## Examining influences of blending *Biome* data on SDM accuracy

Given that the accuracy of SDMs is affected by the amount and quality of data (*Araújo et al., 2019*; *Erickson and Smith, 2023*; *Stockwell and Peterson, 2002*), blending *Biome* data in SDMs may affect the model performances in two possible ways: by increasing the overall amount of data and/or by introducing data with different information than the original data. We analysed to distinguish between these effects. We prepared two different datasets: 'Traditional survey data' and 'Biome + Traditional data'. Then, we separately trained SDMs using these two datasets. We further varied the data size by performing random downsampling, ranging from a minimum of 20 to a maximum of 20,000 records, in order to evaluate its impact on the model. As for the 'Biome + Traditional data' category, the proportion of *Biome* data was kept at 50%. For each condition, we conducted three iterations of training and testing to reduce the impact of random sampling stochasticity. Because the modelling was performed for each species, we obtained BI for each species, amount of records, and dataset (i.e. two datasets consisted of 132 species, each with a maximum of 123 conditions for the amount of records, and the models were replicated three times, resulting in a total of 12,351 individual model runs).

After obtaining BIs for each run, we evaluated the effects of data type (i.e. Biome + Traditional data or Traditional survey data) and species on BI while accounting for the amount of records. For each species and under each amount of records, the mean BI was calculated across the three iterations. Given that BI is a correlation coefficient, we applied the Fisher z-transformation to these BIs to approximate their distribution as a normal distribution. To the transformed BIs, we fitted a generalised linear mixed model that accounted for both the fixed and interaction effects of data type and amount of records. This model accommodated species identity as a random effect. The model was implemented and tested using R packages lme4 (*Bates et al., 2015*) and lmerTest (*Kuznetsova et al., 2017*), respectively.

## Acknowledgements

We are grateful to Jamie M Kass for advice on the construction and evaluation of SDMs. We thank Midzuho Tatsuno, Kazumichi Morishita, Hironori Tanaka, Kotaro Takai and Shuhei Tochino for species identification; Akira Sawada and Yuko Maegawa for their advice on the analysis; Trevor H Booth and Dalelan Anderson for comments on the manuscript. We appreciate the invaluable contributions of Biome users. This research was partly supported by the collaborative research agreement between MU at Kyoto University and Biome Inc from 2020 to 2021.

## Additional information

### Competing interests
Keisuke Atsumi, Yuusuke Nishida: Employed by Biome Inc, but not financially benefit directly from the publication of this paper. Takanori Genroku: CTO of Biome Inc, and an inventor of the species-identification-AI-algorithm JPN patents 6590417 and US patents 11048969, but not financially benefit directly from the publication of this paper. Shogoro Fujiki: CEO of Biome Inc, and an inventor of the species-identification-AI-algorithm JPN patents 6590417 and US patents 11048969, but not financially benefit directly from the publication of this paper. The other authors declare that no competing interests exist.

### Funding
No external funding was received for this work.

## Author contributions

Keisuke Atsumi, Conceptualization, Data curation, Investigation, Visualization, Methodology, Writing – original draft, Writing – review and editing; Yuusuke Nishida, Data curation, Formal analysis, Investigation, Methodology; Masayuki Ushio, Supervision, Methodology; Hirotaka Nishi, Data curation; Takanori Genroku, Conceptualization, Software; Shogoro Fujiki, Conceptualization, Software, Supervision

## Author ORCIDs

Keisuke Atsumi ⓘ https://orcid.org/0000-0002-8206-4977
Masayuki Ushio ⓘ http://orcid.org/0000-0003-4831-7181
Shogoro Fujiki ⓘ https://orcid.org/0000-0002-9778-9532

Reviewer #1 (Public Review): https://doi.org/10.7554/eLife.93694.3.sa1
Author response https://doi.org/10.7554/eLife.93694.3.sa2

## Additional files

### Supplementary files

• Supplementary file 1. Distributions of occurrence records along with environmental variables.
• Supplementary file 2. List of GBIF data doi and literature compiled in occurrence data.
• Supplementary file 3. List of species for constructed species distribution models.
• MDAR checklist

### Data availability

Our analytic code and data are posted on Figshare (https://doi.org/10.6084/m9.figshare.25572462). However, the occurrence data of red-listed species are available upon request for research or application purposes.

The following dataset was generated:

| Author(s) | Year | Dataset title | Dataset URL | Database and Identifier |
|---|---|---|---|---|
| Atsumi K, Nishida Y, Ushio M, Nishi H, Genroku T, Fujiki S | 2024 | Scirpts and data of the article "Boosting biodiversity monitoring using smartphone-driven, rapidly accumulating community-sourced data " | https://figshare.com/articles/dataset/_b_Scripts_and_data_of_the_article_Boosting_biodiversity_monitoring_b_b_using_smartphone-driven_b_b_rapidly_accumulating_b_b_community-sourced_data_b_/25572462 | figshare, 10.6084/m9.figshare.25572462 |

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

# Appendix 1

## Determine the best blend of Traditional survey and *Biome* data

### Methods

In this investigation, we aimed to determine the optimal proportion of *Biome* data within the training dataset of SDMs in order to enhance the accuracy of SDMs. To conduct this assessment, we initially selected a subset of species for which sufficient test data was available (as detailed below).

For each of the selected species, we generated training datasets by combining Traditional survey data with *Biome* data at different proportions: 15, 30, 40, 50, 60, 70, 85, and 100%, referred to as *Biome*-blended datasets.

To compare the accuracy of SDMs, we created and evaluated models using both the Traditional survey dataset and each of the *Biome*-blended datasets. SDMs were created by following the methodology employed in the main analysis. To ensure equitable comparison, we equalised the amount of data in each pair of blended and Traditional survey datasets. This equalisation was achieved by randomly downsampling the larger dataset to match the size of the smaller one.

We assessed the accuracy of the models using the BI, which follows the same methodology as employed in the main analysis. In this specific investigation, we did not control the proportion of *Biome* data within the test data. We selected a set of species for which the test dataset contained at least 50 locations and randomly chose 20 species from each of the seed plants, insects, and birds (see ***Supplementary file 3***).

### Results

The analysis revealed that the relative model accuracy becomes high positive values when training data comprises 50–70% of *Biome* data (***Appendix 1—figure 1***). This indicates that SDM accuracy is substantially enhanced when the training data incorporates 50–70% of *Biome* data. The relative model accuracy remained positive in the 15–70% *Biome*-blended datasets, but decreased to negative values in the 85 and 100% *Biome*-blended datasets (***Appendix 1—figure 1***). This suggests that blending *Biome* data generally enhances the accuracy of SDMs, but it is important to include at least 30% Traditional survey data to maintain accuracy. Based on the high performance observed and simplicity, we selected the 50% *Biome*-blended dataset as the Biome + Traditional data for comparing model accuracy with the Traditional survey data in the main text.

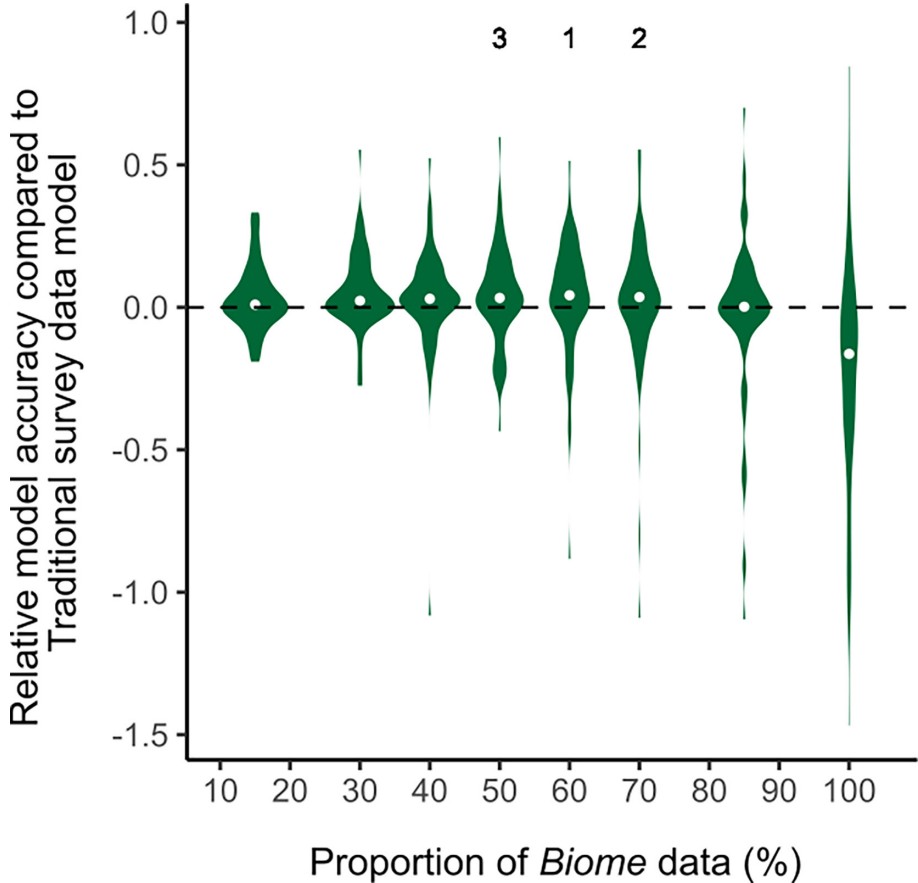

**Appendix 1—figure 1.** The violin plots of relative model accuracy between species distribution models (SDMs) using Biome-blended data and Traditional survey data. The median values are shown as grey dots. The positive relative model accuracy indicates that SDMs that used Biome data outperformed models that used Traditional survey data.

