## [Editor Report · eLife assessment]

This **important** study presents findings of great practical value, offering fresh insights into natural species distributions across Japan. By combining multiple data sources (including those from non-academic sectors, aka citizen scientists), the manuscript also presents a **compelling** new tool that can be used to aid conservation agendas, detect species distribution changes, and testing of ecological theories.

---

## [Referee Report · Reviewer #1 (Public Review)]

Summary:

The study presented by Atsumi et al. is about using smartphone-driven, community-sourced data to enhance biodiversity monitoring. The idea is to leverage the widespread use of smartphones to gather data from the community quickly, contributing to a more comprehensive understanding of biodiversity. The authors discuss the importance of ecosystem services linked to biodiversity and the threats posed by human activities. It emphasizes the need for comprehensive biodiversity data to implement the Kunming-Montreal Global Biodiversity Framework. The 'Biome' mobile app, launched in Japan, uses species identification algorithms and gamification to gather over 6 million observations since 2019. While community-sourced data may have biases, incorporating it into Species Distribution Models (SDMs) improves accuracy, especially for endangered species. The app covers urban-natural gradients uniformly, enhancing traditional survey data biased towards natural areas. Combining these sources provides valuable insights into species distributions for conservation, protected area designation, and ecosystem service assessment.

Strengths:

The use of a smartphone app ('Biome') for community-driven species occurrence data collection represents an innovative and inclusive approach to biodiversity monitoring, leveraging the widespread use of smartphones. The app has successfully accumulated a large volume of species occurrence data since its launch in 2019, showcasing its effectiveness in rapidly gathering information from diverse locations. Despite challenges with certain taxa, the study highlights high species identification accuracy, especially for birds, reptiles, mammals, and amphibians, making the 'Biome' app a reliable tool for species observation. The integration of community-sourced data into Species Distribution Models (SDMs) improves the accuracy of predicting species distributions. This has implications for conservation planning, including the designation of protected areas and assessment of ecosystem services. The rapid accumulation of data and advancements in machine learning methods open up opportunities for conducting time-series analyses, contributing to the understanding of ecosystem stability and interaction strength over time. The study emphasizes the collaborative nature of the platform, fostering collaboration among diverse stakeholders, including local communities, private companies, and government agencies. This inclusive approach is essential for effective biodiversity assessment and decision-making. The platform's engagement with various stakeholders, including local communities, supports biodiversity assessment, management planning, and informed decision-making. Additionally, the app's role in fostering nature-positive awareness in society is highlighted as a significant contribution to creating a sustainable society.

Weaknesses:

While the studies make significant contributions to biodiversity monitoring, they also have some weaknesses. Firstly, relying on smartphone-driven, community-sourced data may introduce spatial and taxonomic biases. The 'Biome' app, for example, showed lower accuracy for certain taxa like seed plants, molluscs, and fishes, potentially impacting the reliability of the gathered data. Furthermore, the effectiveness of Species Distribution Models (SDMs) relies on the assumption that biases in community-sourced data can be adequately accounted for. The unique distribution patterns of the 'Biome' data, covering urban-natural gradients uniformly, might not fully represent the diversity of certain ecosystems, potentially leading to inaccuracies in the models. Moreover, the divergence in data distribution patterns along environmental gradients between 'Biome' data and traditional survey data raises concerns. The app data shows a more uniform distribution across natural-urban gradients, while traditional data is biased towards natural areas. This discrepancy may impact the representation of certain ecosystems and influence the accuracy of Species Distribution Models (SDMs). While the integration of 'Biome' data into SDMs improves accuracy, the study notes that controlling the sampling efforts is crucial. Spatially-biased sampling efforts in community-sourced data need careful consideration, and efforts to control biases are essential for reliable predictions.

---

## [Author Response]

The following is the authors’ response to the original reviews.

**Reviewer #1 (Recommendations For The Authors):**
(1) The modeling process is outlined, but an explanation of why Maxent (Phillips & Dudík, 2008) was chosen for SDMs and why the specified predictor variables were used could provide additional context. This clarity would help readers understand the rationale behind the methodology.

In L.558-571 (*Predictor variables* subsection), we added the explanation about predictor variables as follows:

“Predictors encompass a range of environmental variables recognized to impact species distribution (Table 3): land use (Newbold et al., 2015), climate (bioclim variables (Booth et al., 2014)), vegetation (Abe, 2018), lithology (Ott, 2020) and elevational range (Udy et al., 2021). Additionally, categorical variables representing known biogeographic regions, reflecting geological history, were included. We applied Blakiston's Line —Tsugaru straits dividing the northern and main islands of Japan (i.e., Hokkaido and Honshu islands)— reflecting a significant historical migration barrier for mammals and birds (Dobson, 1994; Saitoh et al., 2015). Due to the distinct fauna (Wepfer et al., 2016; Yamasaki, 2017), we also specified oceanic islands (i.e. Ogasawara and Daito isles) which have never been connected with the Asiatic continents. Continuous environmental variables were transformed into linear, quadratic and hinge feature classes to illustrate nonlinear associations between environments and species occurrence (Phillips et al., 2017). The regularisation multiplier was set at 2.5, falling within the established optimal range of 1.5 to 4 (Elith et al., 2010; MorenoAmat et al., 2015).”

In L.614-618 (*Modelling* subsection), we explain why we chose MaxEnt:

“To model species distributions from presence-only data, several algorithms have been utilised, including generalised additive models, random forest, and neural networks (Norberg et al., 2019; Valavi et al., 2022). In our study, we opted for MaxEnt (Phillips and Dudík, 2008) due to its high estimation accuracy and relatively low computational burden (Valavi et al., 2022).”

(2) While the study outlines a manual reidentification process by experts for wild individuals, it might be beneficial to elaborate on the criteria or expertise level of these experts. This transparency ensures the reliability of the reidentification process. Reply

In L.519-523, we added description about experts as follows:

“These experts have professional backgrounds, serving as a technician at a prefectural research institute (fish), highly-experienced field survey conductors (plants and insects, respectively), a post-doctoral researchers (amphibians and reptiles, and mammals, respectively), and a museum curator (mollusks) specialising in the focal taxa.”

(3) The analysis of the effects of data type (Biome+Traditional data or Traditional survey data) on BI is comprehensive. However, a brief discussion on the potential implications of these effects on the study's overall conclusions could add depth to the interpretation.

We enforced our discussion about the causes and consequences of improved modelling accuracy.

In L.276-282, we argued about the causes:

“Therefore, incorporating *Biome* data could significantly enhance modelling accuracy in urban and suburban landscapes, which are typically underrepresented in traditional survey data. As pseudo-absences are selected based on search effort, our models utilise numerous pseudoabsences from these areas. Consequently, this might lead to better estimation of species absence in such areas, not just presence, resulting in an overall increase in model accuracy across a wider range of species.”

In L.370-387, we argued how improved modelling accuracy may help build naturepositive society as follows:

“By blending data from traditional surveys and communities, we improved the accuracy of species distribution estimates. This enhanced estimation lays the groundwork for more precise subsequent analyses. For instance, estimated distributions will be useful in selecting new protected areas or areas with OECMs (Other Effective area-based Conservation Measures: allowing a wider range of land use as long as biodiversity and ecosystem services are sustained/improved). Using estimated distributions of each species, hotspots of species or evolutionary diverse taxa can be inferred. Such sites will be good candidates for protected areas (Jones et al., 2016) or OECMs (Shiono et al., 2021). Further, estimated distributions can be used as input for spatial conservation prioritisation tools (e.g. Marxan (Ball et al., 2009)).

In our experience, stakeholders—including corporate social responsibility managers and conservation practitioners—often seek the list of species potentially inhabiting their locations. Due to the uncertainty of SDMs and their thresholding into presence/absence, on-site surveys remain essential for assessing biodiversity status. SDMs can make such surveys costeffective by screening important locations for on-site assessment (e.g., Locate phase in TNFD framework) and narrowing down the target species for surveying. Improved estimation through SDMs can mitigate risks associated with their use in society and enable more informed decisionmaking for conservation efforts.”

Following the editorial policy, we have reorganised our supplementary materials as follows:

- Formerly Supplementary File 1 - Remains unchanged.

- Formerly Supplementary File 2 - Transferred into the main text, in the subsection "Filtering suspicious occurrence record in *Biome* data" in the Methods section, and Table 2. Citations remain as Supplementary File 2.

-Formerly Supplementary File 3 - Remains unchanged.

-Formerly Supplementary File 4 - Transferred into "Figure 3—figure supplement 1".

-Formerly Supplementary File 5 - Transferred into Figure 4.

- Formerly Supplementary File 6 - Transferred into the main text, in the subsection "Predictor variables" in the Methods section and Table 3.

- Formerly Supplementary File 7 - Transferred into the main text, in the subsection "Pseudo-absence reflecting search effort" in the Methods section and Figure 5.

- Formerly Supplementary File 8 - Transferred into the main text, in the subsection "Model evaluation" in the Methods section and Figure 6.

- Formerly Supplementary File 9 - Renamed as Supplementary File 4.